# Transcriptomic Analysis Reveals Cu/Zn SODs Acting as Hub Genes of SODs in *Hylocereus undatus* Induced by Trypsin during Storage

**DOI:** 10.3390/antiox9020162

**Published:** 2020-02-17

**Authors:** Xinyue Pang, Xinling Li, Xueru Liu, Luning Cai, Bairu Li, Xin Li

**Affiliations:** 1College of Medical Technology and Engineering, Henan University of Science and Technology, Luoyang 471023, China; pangxy78@haust.edu.cn; 2College of Food and Bioengineering, Henan University of Science and Technology, Luoyang 471023, China180318090357@stu.haust.edu.cn (X.L.); 141416100201@stu.haust.edu.cn (L.C.); 171416120208@stu.haust.edu.cn (B.L.); 3Henan Engineering Research Center of Food Microbiology, Luoyang 471000, China; 4National Demonstration Center for Experimental Food Processing and Safety Education, Luoyang 471000, China

**Keywords:** *Hylocereus undatus* (*H. undatus*), cytoscape, protein-protein interaction (PPI), superoxide dismutase (SOD), trypsin

## Abstract

It has been revealed by us that superoxide scavenging is a new activity of trypsin. In this study, the synergistic mechanisms of trypsin and superoxide dismutases (SODs) were evaluated in *Hylocereus undatus* (pitaya). Trypsin significantly improved the storage quality of *H. undatus*, including weight loss impediment and decrease of cellular injury. The regulatory mechanisms of 16 SOD genes by trypsin were revealed using transcriptomic analysis on *H. undatus*. Results revealed that important physiological metabolisms, such as antioxidant activities or metal ion transport were induced, and defense responses were inhibited by trypsin. Furthermore, the results of protein–protein interaction (PPI) networks showed that besides the entire ROS network, the tiny SODs sub-network was also a scale-free network. Cu/Zn SODs acted as the hub that SODs synergized with trypsin during the storage of *H. undatus.*

## 1. Introduction

*Hylocereus undatus (H. undatus)*, is a species of Cactaceae [1]. Based on the commercial value and outstanding nutritional properties, *H. undatus* is becoming increasingly important to the world [2]. Likes most of fruits, the quality of *H. undatus* decreases with time during storage. Till now, few works have aimed to increase the storage quality of *H. undatus* [3].

The disequilibrium of reactive oxygen species (ROS) production and scavenging will lead to elevated ROS levels, and further induce an increase of cellular injury and finally result in fruit spoilage by the end of storage of fruit [4,5]. As the primary antioxidant enzyme, SODs play important roles in plant physiology [6]. The activities of superoxide dismutase (SOD) have been reported to be induced during fruit spoilage [7], while the regulatory mechanisms among three groups (iron SOD (FSD), copper/zinc SOD (CSD), and manganese SOD (MSD)) during fruit storage have not been elucidated to date.

Trypsin is a widely used protease. It has been confirmed that trypsin significantly impacts on the free radical scavenging activities of flavonoids [8]. Our previous results also showed that trypsin scavenges superoxide anions (O_2_•^−^) and protects cells [9]. 

In the area of phytopathology, transcriptomic analysis has been successful used on lots of plants [10,11,12,13]. However, few of transcriptomic works have been focused on the preservation of fruits. The mechanisms of betalain biosynthesis have been investigated in *H. undatus* [2], and the concision analysis of antioxidant system induced by trypsin has also been investigated in our previous paper [14]. Considering the superoxide scavenging activity of trypsin, the function and synergistic mechanisms of trypsin with SODs impact on the quality of *H. undatus* during storage is an interesting work. The mechanisms of postharvest quality of fruits still need more works. 

Protein–protein interaction (PPI) networks analyses help us to figure out the regulatory mechanisms of proteins [15]. Plugins of Cytoscape, including NetworkAnalyzer, Molecular Complex Detection (MCODE), and cytoHubba, could score and rank the nodes or obtain clusters in the PPI network [16].

In the current work, the regulatory mechanisms of trypsin on the *H. undatus* by cooperating with SODs were investigated. The changes in expression of SODs of *H. undatus* peels were analyzed. Analyses of gene ontology (GO) and KEGG enrichment of ROS or SOD related genes were performed, and the PPI network of ROS related genes and sub-network of SODs and their first neighbors were constructed. The hub genes of SODs regulated by trypsin during storage were further screened by cytoHubba and MCODE of cytoscape.

## 2. Materials and Methods 

### 2.1. Main Materials

*H. undatus* (Vietnam No.1 cultivar) was harvested from the county in Ruyang, Luoyang city in China. Trypsin (bovine, 500 units/mg, crystalline) was purchased from Amersco (Solon, OH, USA). 

### 2.2. H. undatus Treatment 

Forty-five fruits (about 15 cm in diameter) of *H. undatus* were divided into 3 groups, including trypsin, chitosan, and control group. Trypsin, chitosan, or PBS buffer were brushed evenly for 80 s onto the peels [14,17]. The storage conditions of incubator were 25 °C, 85% relative humidity. The indices of fruit quality were detected. The final concentration of trypsin was 2.41 × 10^−6^ mol/L.

### 2.3. Library Construction and Illumina RNA-Sequencing 

Total RNA extraction and transcriptome libraries for RNA-seq of trypsin and control group were performed as reported by us [14]. 

### 2.4. Difference of Gene Expression

To identify the change of expression of ROS or SOD related genes between control and trypsin treated samples, the expression level of each transcript was calculated by the fragments per kilobase of exon per million mapped reads (FRKM) method. Differential expression analysis was performed by EdgeR software (Empirical analysis of Digital Gene Expression in R, http://www.bioconductor.org/packages/2.12/bioc/html/edgeR.html) from the R statistical package. The false discovery rate (FDR) was used to adjust the resulting *p*-values using the Benjamini and Hochberg approach [18,19].

### 2.5. GO and KEGG Enrichment Analyses 

GO and Kyoto Encyclopedia of Genes and Genomes pathway (KEGG) information were enriched as described by Candar-Cakir et al. [20]. The enriched GO terms were shown with bar charts and DAGs (directed acyclic hierarchical graph) [20].

### 2.6. Gene Expression Analysis by Reverse Transcription-qPCR

Reverse transcription-qPCR was performed as described by Yang et al. [21]. The information of primer was shown in Table 6. Gene expression was normalized by internal control, β-actin. The relative copy numbers of the genes were calculated by the 2^−∆∆Ct^ method [11,21].

### 2.7. Protein-Protein Interaction (PPI) Analyse

#### 2.7.1. PPI Network Generation

Proteins were exported from the cloud platform of I-Sanger and loaded into cytoscape to construct a network. The interactions between ROS or SOD related proteins regulated by trypsin were shown by the nodes and edges in the networks of ROS, the first neighbors of SODs or SODs of *H. undatus* using cytoscape [22].

#### 2.7.2. Network Topological Parameters

NetworkAnalyzer showed the basic network parameters in this study [22]. Here, the edges in the networks constructed in the current study were calculated as undirected. The power law curve was formed as *y* = β*x*^α^ [23].

#### 2.7.3. Module

The clusters of the entire ROS related network were identified by MCODE [24]. The edges in networks were set as directed for MCODE or next hub node analysis.

#### 2.7.4. Analysis of Hub Nodes in the PPI Network

The cytoHubba plugin of cytoscape was used to identify the high degree nodes [25] by 11 topological analysis methods [26].

### 2.8. Weight Loss Rate

The rates of weight loss of samples were obtained using 3 parallel experiments and were recorded at 72 and 168 h (25 °C). 

### 2.9. Quantification of Lipid Peroxides

Malondialdehyde (MDA) contents were measured as described by Zhou et al. [27].

### 2.10. Determination of ROS of H. undatus Peel

The production of O_2_•^−^ or hydrogen peroxide (H_2_O_2_) accumulation was determined as described by Schneider and Schlegel [14,28] or Li and Imlay in 2018 [29], respectively.

### 2.11. Statistical Analyses 

The statistical analyses were performed by SPSS statistical software package (11.0.1) (15 November 2001, SPSS Inc., Chicago, IL, USA). The differences between samples were analyzed by paired sample *t*-test. Significant or highly significant difference was evaluated by *p* < 0.05 or *p* < 0.01, respectively.

## 3. Results

### 3.1. Effect on Fruit Quality of H. undatus

The appearance of three groups of *H. undatus* was bright and fresh at 0 h (Figure 1A–C). After 168 h of storage, the fruit of control group were entirely decayed (Figure 1D). The peel of fruit of trypsin group was preserved well and similar to that at the beginning of storage (Figure 1E). The decay of chitosan treated pitaya was much less than the control, while there were multiple rotting spots on the fruits (Figure 1F).

The weight loss of each group showed a significant increasing with increasing storage time (Figure 1G). The weight loss rates were 1.15%, 0.78%, or 0.84% in control, trypsin or chitosan group (Figure 1G). The difference between the two preservative groups (trypsin and chitosan) was not significant. While both trypsin and chitosan group exhibited a highly significant difference with the control group at 72 or 168 h (*p* < 0.01).

### 3.2. Impact on the Cell Injury

In control group, results showed that the MDA contents sharply increased by 305% after 168 h of storage (Figure 1H). The increase of MDA was entirely inhibited by trypsin. The effect of trypsin was much better than that of chitosan (*p* < 0.05) (Figure 1H). The difference between the control group and the trypsin or chitosan group was highly significant at 168 h (*p* < 0.01). The result suggested that trypsin can significantly impede the formation of MDA contents, which represents the lipid peroxidation of cellular membrane. In other words, trypsin significantly inhibits the cell injury by excess ROS during storage.

### 3.3. Impact on the ROS of H. undatus

As shown in Figure 1I,J, the rate of O_2_•^−^ production and H_2_O_2_ levels in the fruits of the control group increased with storage. Either trypsin or chitosan inhibited the accumulation of ROS (*p* < 0.01) (Figure 1I,J). Trypsin completely inhibited the accumulation of O_2_•^−^ and H_2_O_2_.

### 3.4. Transcriptomic Analyses

#### 3.4.1. Differentially Expressed Genes

Since there was still no reference genome for *H. undatus*, transcripts and unigenes were blasted against 6 major databases (Appendix A). A total of 30,222 (34.81%) unigenes were annotated (*E* value < 10^−6^) (Appendix A).

#### 3.4.2. Functional Annotation and Analyses

Sixteen superoxide related genes were screened, including four up- and five down-regulated genes (Appendix A).

Based on clusters of orthologous groups of proteins (COG) classifications, seven unigenes were assigned into only one functional category, inorganic ion transport and metabolism, which belongs to the metabolism type. All 16 superoxide related unigenes were mainly classified into three categories of gene ontology (GO) i.e., biological process (BP), molecular function (MF), and cellular component (CC) on level 2 (Appendix A). The main functions were gathered in catalytic activity (12 unigenes, 75%), binding (13 unigenes, 81.25%), and antioxidant activity (8 unigenes, 50%) of MF classification. As for the BP, they were focused on cellular process (6 unigenes, 37.50%), metabolic process (5 unigenes, 31.25%), and localization (5 unigenes, 31.25%) (Appendix A). In the cellular component, cell part (3 unigenes, 18.75%) and cell (3 unigenes, 18.75%) were the major parts (Appendix A). 

Sixteen unigenes were assigned to 5 KEGG pathways (Appendix A). SOD was annotated in the hydrogen peroxide metabolism (PTS1 type) belonging to the antioxidant system which is represented in the peroxisome biogenesis pathway (map04146) (Appendix A). Serine-protein kinase ATM (EC:2.7.11.1) play key roles in the recognition part of pathway homologous recombination (map03440) (Appendix A). The signal recognition particle receptor subunit beta (SRPRB) was the key protein in the sec dependent pathway of the protein export pathway (map03060) (Appendix A). Nonsense-mediated mRNA decay protein 3 (NMD3) played key roles (Appendix A) either in pathway RNA transport (map03013) or ribosome biogenesis in eukaryotes (map03008). 

#### 3.4.3. GO Enrichment Analyses

GO-based enrichment analysis showed the biological functions of the patterns up- or down-regulated by treatment with trypsin. The top 10 significantly enriched GO terms in the two expression patterns are shown in Table 1 (*p*-value < 0.01). The up-regulated pattern was enriched with GO terms of superoxide metabolic process (GO:0006801), metal ion transport (GO:0030001), superoxide dismutase activity (GO:0004784) and oxidoreductase activity, acting on superoxide radicals as acceptor (GO: 0016721). On the other hand, besides GO: 0004784 and GO:0016721, antioxidant activity (GO:0016209), oxidoreductase activity (GO:0016491), cellular response to UV-B (GO:0071493), or cellular response to ozone (GO:0071457), were down-regulated (Table 1, Appendix A).

The pathways related to superoxide involved in trypsin regulation can be summarized in a DAG (Figure 2). For example, the GO biological process “superoxide metabolic process (GO:0006801)” is a child of one term: “reactive oxygen species metabolic process (GO:0072593)”. The GO biological process “metal ion transport (GO:0030001)” is a child of one term: “cation transport (GO:0006812)”. When the GO enrichment was analyzed separated by two patterns, more information exhibited. Cellular responses including to ozone (GO:0071457), to high light intensity (GO:0071486) or to UV-B (GO:0071493) were down-regulated (Appendix A). Besides “superoxide metabolic process (GO:0006801)”, “metal ion transport (GO:0030001)” was shown in up-regulated “biological process (GO:0008150)” of GO process (Appendix A). Furthermore, “superoxide dismutase activity (GO:0004784)”, and “metal ion binding (GO:0046872)” was shown in up-regulated “molecular function (GO:0003674)” of GO process (Appendix A). 

#### 3.4.4. KEGG Enrichment Analyses

Table 2 showed all of the KEGG pathway enrichment regulated by trypsin. No matter up- or down-regulated genes were enriched in the peroxisome pathway (map04146) (Appendix A). Protein export pathway (map03060) and homologous recombination (map03440) were up regulated (Appendix A), while RNA transport (map03013) and ribosome biogenesis in eukaryotes (map03008) were down regulated (Appendix A). The statistical significance of these pathways is listed in Table 2.

### 3.5. PPI Network Analysis

#### 3.5.1. PPI Networks of SODs

Totally, we obtained 16 unigenes of SODs (four up- and five down-regulated, Appendix A) involved in 1027 ROS related genes (FC > 0.8), including 432 up-regulated genes and 377 down-regulated genes (Appendix A). Seven of the 16 SODs constructed a sub-network (Figure 3 and Appendix A). 

The PPI subnetwork of total ROS genes was composed of 447 nodes and the first 3000 edges (Figure 4 and Appendix A). The cytoscape plugin MCODE was layered on and obtained 17 clusters (Appendix A). Nodes belonging to the top five clusters were labeled by different colors (Figure 4). Among these, six of the SODs were highlighted by blue color in cluster 1 and cluster 3 of Figure 5. The CCS (copper chaperone for SOD) was not represented in the clusters of the whole ROS network. 

The sub-network of the first neighbors of SODs were further screened from the PPI network of ROS (Figure 6). The first neighbors of SODs PPI sub-network contained 65 nodes and 693 edges, including 38 up-regulated and 23 down-regulated proteins (Appendix A). Analyzed by MCODE, 23, 14, and 3 nodes were gathered in three clusters, respectively (Appendix A). Six SODs were gathered in cluster1 of the SOD first neighbor sub-network (Appendix A), and CCS was still absent (Appendix A). 

Furthermore, the sub-network of SODs was constructed on I-Sanger cloud platform and figured by cytoscape (Figure 3A). Two FSDs (Fe-SOD) were up-regulated, while two of three CSDs (Cu/Zn SOD) and MSD1 (Mn-SOD) were down-regulated. Based on 11 ranked methods including Maximal Clique Centrality (MCC), Density of Maximum Neighborhood Component (DMNC), Maximum Neighborhood Component (MNC), Degree, Closeness, Betweenness, Radiality, EcCentricity, Stress, Clustering Coefficient, EPC, and BottleNeck in cytoHubba, 7 SODs were ranked (Appendix A). Results showed that seven SODs were ranked into three classes by most of these methods. Three CSDs were classified into the first class. Two FSDs and MSD1 were classified into the second class. CCS was classified as the last one (Figure 3B and Appendix A). 

#### 3.5.2. Topological Properties of Networks. 

The node degree distributions of the total ROS related genes network followed power law fit distributions (*R*^2^ = 0.816) (Appendix A and Table 3). Other subnetwork topological parameters, such as clustering coefficient, and network density were shown in Table 3. 

The correlation of the first neighbor of SODs sub-network was decreased to −0.006 (*R*^2^ = 0.000) with the decreasing of nodes (Appendix A and Table 3). Although the seven SODs constructed a tiny sub-network, either the Clustering coefficient or the Network density was much higher (0.914 and 0.857, respectively) than that of the first neighbor of SODs sub-network (0.598 and 0.333, respectively) or even total ROS network (0.384 and 0.030, respectively). The correlation of the SODs sub-network was 0.924 (*R*^2^ = 0.936) (Appendix A and Table 3).

### 3.6. Accuracy of the RNA-Seq Data Verification by RT-qPCR

The changes of expression of seven SOD genes were checked by RT-qPCR (Figure 7). The full information of these seven SOD genes were shown in Appendix A.

## 4. Discussion

Trypsin significantly impede the loss of water, dehydration and improved fruit appearance quality. The level of malondialdehyde (MDA) is known to be an effective index for cell membrane lipid peroxidation [30]. Trypsin significantly decreased the MDA contents, leading to significantly reduced cell damage from lipid peroxidation.

During the course of maturity or decline of *H. undatus*, the accumulation of excess O_2_•^−^ and H_2_O_2_ can cause membrane damage [31,32]. As anticipated, either trypsin or chitosan inhibited the accumulation of ROS. The inhibition of trypsin, the novel superoxide scavenger, was entirely, especially on O_2_•^−^.

This study explored several intriguing questions about the effect of trypsin treatment on SOD mediated regulatory responses. How does trypsin perform ROS regulation? As a novel scavenger of O_2_•^−^, does trypsin impact the expression of SODs? Transcriptomic analysis was used to investigate the change in expression of SODs and identify the key SOD genes involved in ROS metabolism through trypsin treatment.

Showed by transcriptomic data, 16 superoxide related genes were screened, including four up-regulated genes and five down-regulated genes. The number of SOD unigenes are different in plants reported. For example, there are 7 SODS in *Arabidopsis* [33], 10 SODs in *Cakile maritima* [34], and 4 SODs in pepper [35]. A total of 29 and 18 SOD genes have been identified in *Brassica juncea* and *Brassica rapa*, respectively [36]. The evolutionary relationship study among plants need more works.

All of these 16 unigenes were mainly classified into three categories of GO, including BP, MF, and CC on level 2. The DAGs indicated that the superoxide metabolism and metal ion transport are key processes of SOD related regulatory mechanisms by trypsin during *H. undatus* storage. Important physiology metabolisms, such as antioxidant activities or metal ion transport, were induced, while defense responses were impeded by trypsin (Appendix A). The results of GO enrichment suggested that trypsin exhibited the protection of cell injury of *H. undatus* during storage.

In addition, the functional involvement of the pathways in the trypsin responsive patterns were shown by KEGG enrichment analyses. With trypsin treatment, different SODs in the peroxisome pathway were regulated by up or down patterns (Appendix A). Protein export pathway (map03060), homologous recombination (map03440), RNA transport (map03013) and Ribosome biogenesis in eukaryotes (map03008) were involved in the regulation of trypsin in *H. undatus*.

It is critical to explore the status of SODs in the entire ROS network. The cytoscape plugin “MCODE” was layered on the PPI of total ROS genes to illustrate the attribution of SODs in sub-networks. Seventeen clusters were obtained. The absence of CCS implies that it is a specific assistant protein of SOD and works independently.

To further investigate the mechanisms of SODs, the sub-network of the first neighbors of SODs were screened from the PPI network of ROS. Besides the 7 SODs, 58 proteins were involved as the first neighbors of SODs PPI sub-network (Figure 6 and Appendix A). This PPI sub-network indicated that trypsin deeply interfered in the PPI network in *H. undatus*, as SODs interacted with lots of proteins to enlarge the biological effects as compared with the PPI of total ROS genes.

Finally, the DAG of 7 SODs indicated that CSDs acted as hubs, and either CSDs or MSD1 were down-regulated due to their synergistic effect with trypsin, which is also a Cu ion catalyzed superoxide scavenger. Overall, trypsin in combination with CSDs induced higher FSDs activities, cooperated by CCS and MSD1 (Figure 3A).

The fact that CSDs act as hub SODs or that the copper chaperone for SOD (CCS) closely interact with SODs, were strongly consisted with those in our previous reports where it was identified that Cu ion was involved in the mechanisms of trypsin by dynamical analysis using ProDy [8,37].

It is generally acknowledged that the scale-free network could be judged according to the node degree distribution of a PPI network [22,38]. The node degree distributions of the total ROS related genes network followed power law fit distributions. This indicated that the ROS PPI network is a true complex biological scale-free network.

Considering the sub-network of the first neighbors of SODs was constructed by all of the first neighbors of the SODs, the entire network exhibited high homogeneity. In addition, the 38 up-regulated (58.5%) and 23 down-regulated (35.4%) nodes in this sub-network indicated that nodes closer to the SODs were much more regulated by trypsin, compared with those elsewhere in the total ROS network (26.8% up- and 33.5% down-regulated). As anticipated, the correlation of the first neighbor of SODs sub-network was decreased to −0.006 (*R*^2^ = 0.000) with the decreasing of nodes (Appendix A and Table 3). So, it was characterized as a non-scale-free network.

Although the seven SODs constructed a tiny sub-network, likely because of an eminent gradient difference across these seven nodes, either the clustering coefficient or the network density was much higher than that of the first neighbors of SODs sub-network or even the total ROS network. All of the above results indicated that the 3 CSDs were hubs of the SODs. Moreover, considering the networks constructed in the current study, it was suggested that the criterion of a scale-free network might not be dependent upon the number of nodes and that the biological networks are not always scale-free networks.

The accuracy of the RNA-Seq data was confirmed by RT-qPCR. Expression changes of seven SOD genes were consistent with the results of RNA-Seq.

## 5. Conclusions

The trypsin treatment significantly reduced the accumulation of endogenous ROS, including O_2_•^−^ and H_2_O_2_, impeded the cell injury, and improved the storage quality of *H. undatus*. Transcriptomic analysis identified seven SOD genes regulated by trypsin in *H. undatus*. Results revealed that important physiological metabolisms, such as antioxidant activities or metal ion transport, were induced, and defense responses were inhibited by trypsin. PPI network analysis suggested that the Cu/Zn SODs act as hub SODs synergized with trypsin, and induced higher FSDs activities, cooperated by CCS and MSD1 during storage of *H. undatus*. As a highly efficient, safe and economical antioxidant, trypsin might be used widely as a new bio-preservative.

## Figures and Tables

**Figure 1 antioxidants-09-00162-f001:**
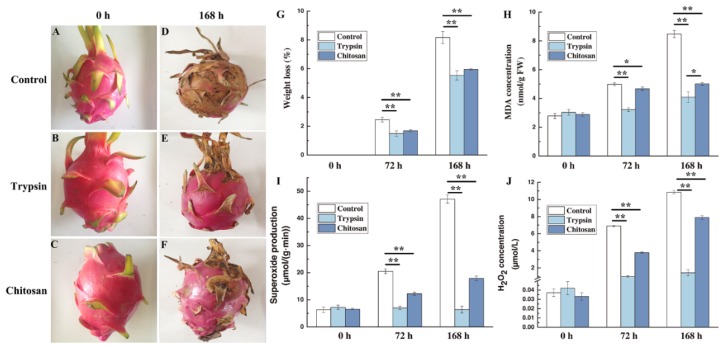
Effect of trypsin treatment on the storage quality or ROS levels of *H. undatus*. Values are the mean ± SE of triplicate samples. (**A**–**F**) Fruits of *H. undatus* stored at 25 °C for 168 h with and without trypsin or chitosan treatment. (**A**,**D**) Control fruits, (**B**,**E**) trypsin-treated fruits, (**C**,**F**) chitosan-treated fruits, 0 or 168 h represent that fruits were stored for 0 or 168 h, respectively, (**G**) weight loss of *H. undatus* fruit stored with or without trypsin or chitosan for 72 or 168 h, (**H**) malondialdehyde MDA contents of *H. undatus* fruit stored with or without trypsin or chitosan for 72 or 168 h, (**I**,**J**) superoxide anion production rates, or H_2_O_2_ concentrations in *H. undatus* peel with or without trypsin or chitosan for 72 or 168 h. * represented significant difference (*p* < 0.05); ** represented highly significant difference (*p* < 0.01).

**Figure 2 antioxidants-09-00162-f002:**
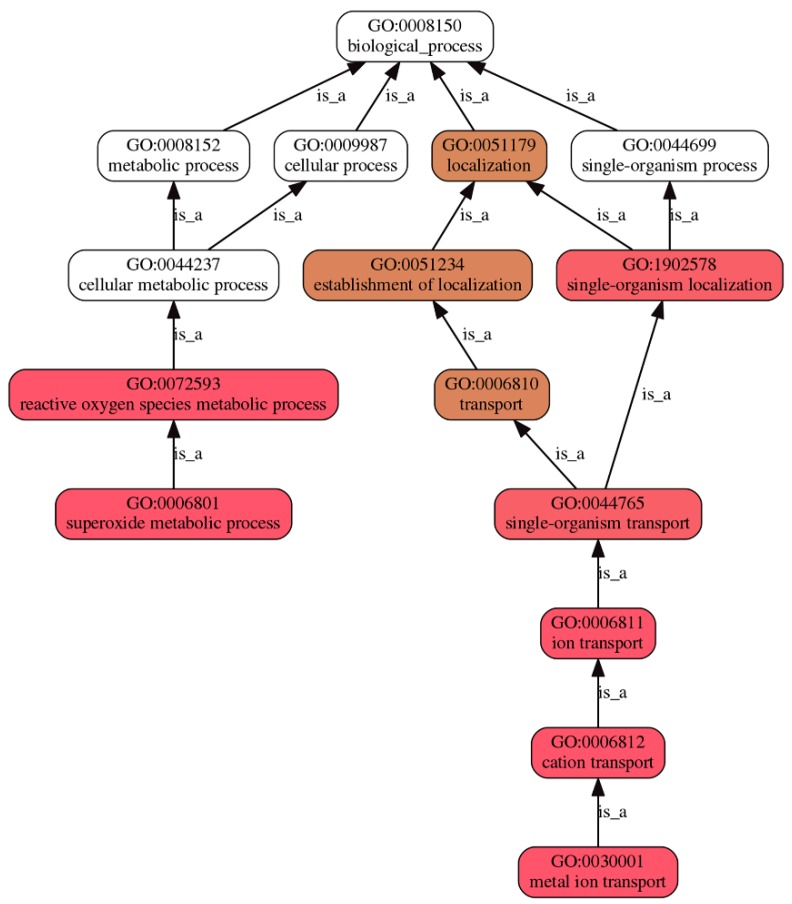
Directed acyclic hierarchical graph (DAG) of gene ontology (GO) terms regulated by trypsin. The figure illustrates a subset of the molecular function DAG for superoxide metabolic process (GO:0006801) and metal ion transport (GO:0030001). The ancestors of GO:0006801 or GO:0030001 are highlighted back to the root of the biological process (GO:0008150) ontology via arrows.

**Figure 3 antioxidants-09-00162-f003:**
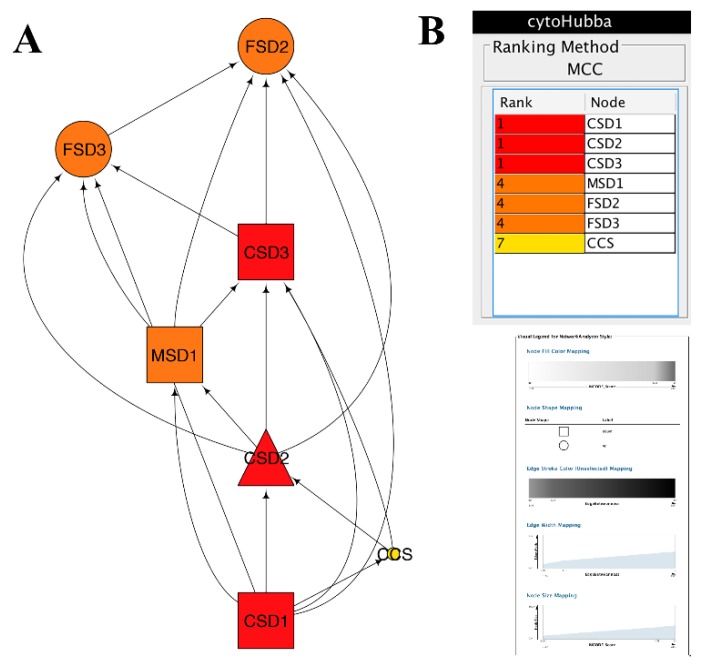
SODs protein–protein interaction PPI sub-networks induced by trypsin were constructed by cytoscape software. Rectangle nodes represented proteins encoded by downregulated genes, while round nodes represented proteins encoded by upregulated genes. The CSD2 without significantly differential expression was represented as triangle-shaped nodes. The nodes were ranked and colored by cytoHubba.

**Figure 4 antioxidants-09-00162-f004:**
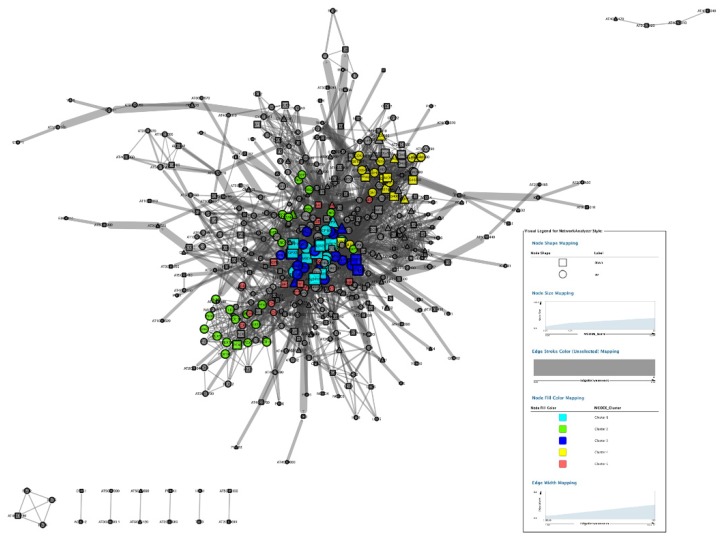
ROS related PPI networks induced by trypsin were constructed by cytoscape software. Rectangle nodes represented proteins encoded by downregulated genes, while round nodes represented proteins encoded by upregulated genes. The other interacting proteins without significantly differential expression were represented as triangle-shaped nodes. Top 5 clusters calculated by MCODE were colored as shown in the legend.

**Figure 5 antioxidants-09-00162-f005:**
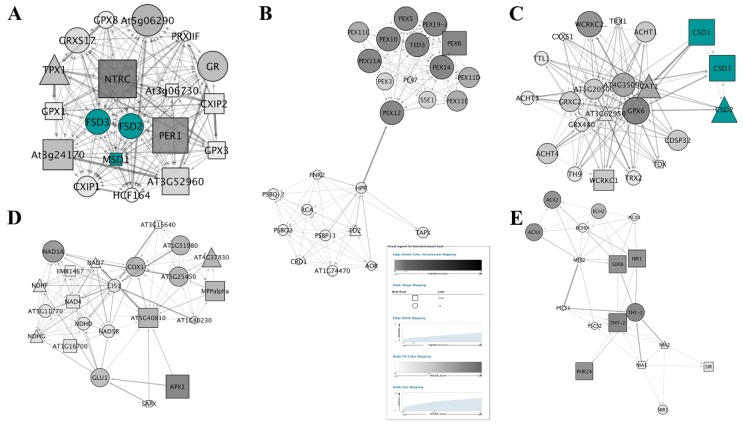
Top 5 clusters of ROS related PPI network calculated by MCODE. (**A**–**E**) represents cluster 1–5, respectively. SODs were colored in (**A**,**C**).

**Figure 6 antioxidants-09-00162-f006:**
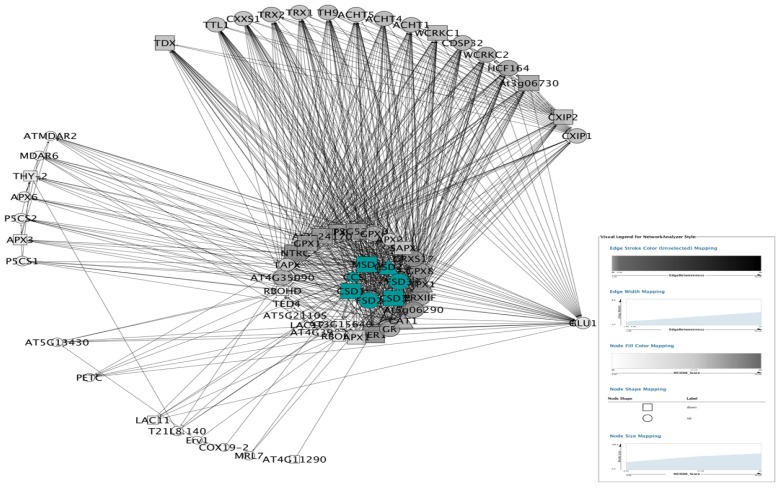
The first-neighbors of SODs PPI sub-networks induced by trypsin were constructed by cytoscape software. Nodes were arranged as attribute circle layout according to the neighborhood connectivity of selected nodes.

**Figure 7 antioxidants-09-00162-f007:**
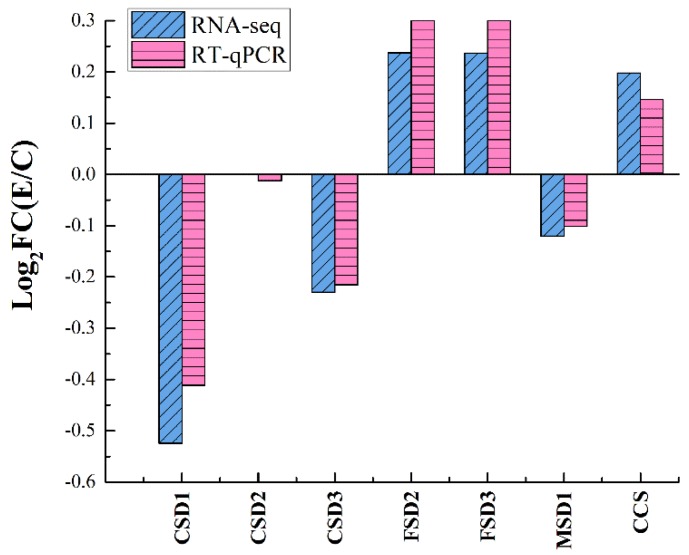
RNA-seq and RT-qPCR determination of 7 SOD genes of *H. undatus* peel at 168 h of storage.

**Table 1 antioxidants-09-00162-t001:** Top 10 GO enrichment terms related to HuSODs (*p* < 0.01) by trypsin.

Pattern	Num	GO ID	Term Type	Description	*p*-Value *	FDR ^a^
**Up-regulation**	1	GO:0006801	BP	superoxide metabolic process	0.002383	1
	1	GO:0030001	BP	metal ion transport	0.009766	1
	1	GO:0004784	MF	superoxide dismutase activity	0.001004	1
	1	GO:0016721	MF	oxidoreductase activity, acting on superoxide radicals as acceptor	0.001004	1
**Down-regulation**	3	GO:0004784	MF	superoxide dismutase activity	3.3214 × 10^−10^	9.1654 × 10^−7^
	3	GO:0016721	MF	oxidoreductase activity, acting on superoxide radicals as acceptor	3.3214 × 10^−10^	9.1654 × 10^−7^
	3	GO:0016209	MF	antioxidant activity	1.2739 × 10^−6^	0.002344
	4	GO:0016491	MF	oxidoreductase activity	2.9523 × 10^−5^	0.04073
	1	GO:0071493	BP	cellular response to UV-B	0.0002510	0.1732
	1	GO:0071457	BP	cellular response to ozone	0.0002510	0.1732

* *p*-values were calculated using Fischer‘s test. ^a^ FDR corrections were calculated using the Benjamini–Hochberg procedure.

**Table 2 antioxidants-09-00162-t002:** Pathways related to superoxide dismutases (SODs) enriched by trypsin.

Pattern	Num	KO ID	Term	*p*-Value *	FDR ^a^
Up-regulation	2	map04146	Peroxisome	0.00068	0.0020
	1	map03060	Protein export	0.023	0.035
	1	map03440	Homologous recombination	0.039	0.039
Down-regulation	3	map04146	Peroxisome	4.84 × 10^−6^	1.45 × 10^−5^
	1	map03008	Ribosome biogenesis in eukaryotes	0.058	0.087
	1	map03013	RNA transport	0.097	0.097

* *p*-values were calculated using Fischer’s test. ^a^ FDR corrections were calculated using the Benjamini–Hochberg procedure.

**Table 3 antioxidants-09-00162-t003:** Topological parameters of the total ROS related genes, a first neighbor of SODs and 7 SODs PPI networks of *H. undatus* regulated by trypsin.

PPI Subnetwork	y = βxα	R^2^	Correlation	Clustering Coefficient	Network Centralization	Network Density	Total Num. of Nodes	Num. of Up-Regulated Nodes	Num. of Down-Regulated Nodes	Characteristic Path Length
Total ROS	*Y* = 109.39*x*^−1.036^	0.816	0.873	0.384	0.254	0.030	447	120	150	3.285
First neighbors of SODs	*Y* = 1.727 *x* ^0.004^	0.000	−0.006	0.598	0.414	0.333	65	38	23	1.758
7 SODs	*Y* = 0.161 *x* ^1.708^	0.936	0.924	0.914	0.200	0.857	7	3	3	1.143

## Data Availability

Raw sequence data from this study have been submitted to the NCBI sequence read archive under the BioProject accession [PRJNA509494] and available at the following link: https://trace.ncbi.nlm.nih.gov/Traces/sra_sub/sub.cgi?acc=SRP173572.

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
