# Peer review of "Transcriptomic Analysis Reveals Cu/Zn SODs Acting as Hub Genes of SODs in Hylocereus undatus Induced by Trypsin during Storage"

_antioxidants, 2020, doi:10.3390/antiox9020162_

Round 1

Reviewer 1 Report

The manuscript by Li and co-workers deals with the synergistic mechanisms of trypsin and superoxide dismutases in Hylocereus undatus.

This Referee has the following mostly serious problems:

There is a recently published paper of the authors „Food Funct., 2019, 10, 8116”, which is the antecedent of the current manuscript, however, it was not cited. It is not really clear to the reviewer, what is the novelty of this manuscript compared to the previous one. It should be clarify. In this ms the storage time was written as 168 h, and in the previous paper was mentioned as 7 days, which are the same.

The importance of this study is also missing. Furthermore, the Referee didn’t get answer from the manuscript about the following questions.

Has there been such research on this plant before?

Are there any other plants, which showed similar results?

It would be advisable to complete the manuscript with these explanations.

In conclusion, this manuscript requires a major revision to be able to be considered further.

Author Response

List of Responses

Dear editors and reviewers,

We have made some corrections based on the comments of you. I have highlighted the changes I make in the manuscript by using the track changes mode in Word. The responses to the comments of you are as follows:

REVIEWER REPORT(S): 
Reviewer 1

The manuscript by Li and co-workers deals with the synergistic mechanisms of trypsin and superoxide dismutases in Hylocereus undatus.

This Referee has the following mostly serious problems:

There is a recently published paper of the authors „Food Funct., 2019, 10, 8116”, which is the antecedent of the current manuscript, however, it was not cited. It is not really clear to the reviewer, what is the novelty of this manuscript compared to the previous one. It should be clarify. In this ms the storage time was written as 168 h, and in the previous paper was mentioned as 7 days, which are the same.

Response: The paper published on “Food Funct” has been cited as reference [13], which was the previous reference [16]. I am sorry that the novelty of this manuscript was not described clear. The noveltieshave been listed above and complemented in the introduction. As for the 168 h, I prefer to use 168 h in the current manuscript, which is the different expression of the time “7 days”.

The importance of this study is also missing. Furthermore, the Referee didn’t get answer from the manuscript about the following questions.

Has there been such research on this plant before?

Response: Although the hub genes involved in the betalain biosynthesis have been screened investigated in H. undatus[2], the concision analysis of antioxidant system induced by trypsin has been investigated in our previous paper [13]. The function and synergistic mechanisms of trypsin with SODs impact on the quality of H. undatusduring storage has not been shown yet. The regulatory mechanisms of postharvest quality of fruits still need more works.

Are there any other plants, which showed similar results?

Response: There are lots of paper of transcriptomic analysis on plants,especially in the area of phytopathology. However, the research on the impact of trypsin on plant is firstly reported by us.

It would be advisable to complete the manuscript with these explanations.

Response: These contents have been complemented in the text.

In conclusion, this manuscript requires a major revision to be able to be considered further.

Response: Thanks for your kindly comments. This manuscript has been improved according to your recommendation.

If you have any question on this manuscript, please no hesitate to contact me at any time. Thank you very much!

Prof. Xin Li

College of Food and Bioengineering

Henan University of Science and Technology

Reviewer 2 Report

The article by Pang, Li et al. is an interesting study, showing the effect of treatment of fruit by trypsine or chitosan on the fruit preservation over time. However, the description of the experiments is not precise enough to allow the reader to understand exactly what was done and then evaluate the impact of the results. I am not fully familiar with the GO and transcriptomic procedure but the scope of comparison and the conclusion should be improved.

Some general questions are raised (Q1, Q2, Q3) :
Q1: How can the brushing only on the surface could affect the whole fruit? Is there a diffusion within the fruit?
Q2: Is there a control with no brushing? Indeed, the brushing (wtih PBS) could impair the surface protection of the fruit and could favor degradation.
Q3: Where the size of the fruit calibrated? Indeed smaller fruits have a higher ratio surface/volume and could loose water at a higher rate.

Some more specific questions about the description of the experiments are also to be answered (Q4, Q5, Q6):
Q4: On what kind of sample are the MDA and superoxide assay performed: is that on the whole fruit or on the peel and if so what was the thickness chosen?
Q5: Superoxide is an unstable anion. Indeed it cannot accumulate in biological samples: how can there be an increasing concentration as it should disappear very fast in a biological sample? It may display a higher rate of production but certainly not an actual accumulation in tissues. This as to be better worded and commented on from the experimental results obtained. Especially, a better description of the assay, on the specific sample, should be given and not only a general reference in another context.
Q6: How many independent samples were studied?

The study of the up-regulation and down-regulation is interesting. But it is not very clear why SOD is overexpressed: what is the rationale for a link between treatment with trypsine and SOD over-expression? They mention that trypsine is synergized the SOD hub through CuZnSOD, but how does that work on the biochemical point of view? What about other important anti-oxidant proteins such as catalase of GSH peroxidase? What is the differential between trypsine and chitosan treatment?

Overall, the article needs to be greatly improved before it could be considered for a publication.

Author Response

List of Responses

Dear editors and reviewers,

We have made some corrections based on the comments of you. I have highlighted the changes I make in the manuscript by using the track changes mode in Word. The responses to the comments of you are as follows:

REVIEWER REPORT(S): 

Reviewer 2

The article by Pang, Li et al. is an interesting study, showing the effect of treatment of fruit by trypsine or chitosan on the fruit preservation over time. However, the description of the experiments is not precise enough to allow the reader to understand exactly what was done and then evaluate the impact of the results. I am not fully familiar with the GO and transcriptomic procedure but the scope of comparison and the conclusion should be improved.

Some general questions are raised (Q1, Q2, Q3) :
Q1: How can the brushing only on the surface could affect the whole fruit? Is there a diffusion within the fruit? 

Response: Yes, your question is really valuable. It has been confirmed that the trypsin is a novel superoxide scavenger and effective preservative. While the recognition mechanism of trypsin is still unclear. The receptor of trypsin is still unknown. It is hypothesized that trypsin activates the hormone signal pathway to regulate endogenous antioxidant system of H. undatus. The signals might be transmitted and expanded through the communication between cells.

Q2: Is there a control with no brushing? Indeed, the brushing (wtih PBS) could impair the surface protection of the fruit and could favor degradation. 

Response: Sure, the PBS treatment will impact on the fruit quality. At the beginning of the experiment, indexes of two control groups with or without PBS were detected. Trypsin group has significant difference with both control groups. While, the transcriptomic sequencing is still too expensive to detect all of these control samples. Furthermore, the comparation between trypsin dissolved in PBS and control group with PBS just shows the effect of trypsin. In the future, we are going to analysis the difference between fruit with and without PBS.

Q3: Where the size of the fruit calibrated? Indeed smaller fruits have a higher ratio surface/volume and could loose water at a higher rate.

Response: Sure, the size of fruit is an important parameter. Fruit without mechanical damage and with a uniform color, size and number of scales was chosen for the study. The fruit used in our work was about 15 cm in diameter. It has been added in the “Materials and Methods” section.

Some more specific questions about the description of the experiments are also to be answered (Q4, Q5, Q6):
Q4: On what kind of sample are the MDA and superoxide assay performed: is that on the whole fruit or on the peel and if so what was the thickness chosen? 

Response: All of the samples used in our experiments were taken from peels. The thickness was about 3 mm.

Q5: Superoxide is an unstable anion. Indeed it cannot accumulate in biological samples: how can there be an increasing concentration as it should disappear very fast in a biological sample? It may display a higher rate of production but certainly not an actual accumulation in tissues. This as to be better worded and commented on from the experimental results obtained. Especially, a better description of the assay, on the specific sample, should be given and not only a general reference in another context.

Response:Sure, superoxide is really unstable. The life of it is so short. So, we described the superoxide and H2O2by production rate (μmol/g. min) and concentration (μmol/L), respectively. We have focus on the ROS production and scavenging for more than ten years. Electron Spin Resonance (ESR) is a better method to catch the signal of superoxide by DMPO, Tiron, etc. (Li et al., 2009; 2017; 2018). While, the method described by Schneider and Schlegel is a classic method to detect the superoxide production. Especially in plant, the superoxide production is much higher than that in bacteria. So, this method is suitable for the current manuscript. Moreover, the superoxide detection is the same to that in our previous work published on Food & Function, 2019. The citation has been added.

Our papers of superoxide determination by ESR is listed as following:

1) Xin Li, et al. Trypsin binding with copper ions scavenges superoxide: Molecular dynamics-based mechanism investigation. International Journal of Environmental Research and Public Health. 2018, 15 (1) :139-152. doi:  10.3390/ijerph15010139.

2) Xin Li, et al. Trypsin slows the ageing of mice due to its novel superoxide scavenging activity. Applied Biochemistry and Biotechnology. 2017, 181(4): 1549-1560. DOI: 10.1007/s12010-016-2301-7.

……

3) Xin Li, et al. Extracellular superoxide anion production contributes to virulence of Xanthomonas oryzae pv.oryzae. Canadian Journal of Microbiology. 2009, 55(2): 110-116. DOI: 10.1139/W08-112

Q6: How many independent samples were studied?

Response: The experiments were performed using 3 replicates (n = 6).

The study of the up-regulation and down-regulation is interesting. But it is not very clear why SOD is overexpressed: what is the rationale for a link between treatment with trypsine and SOD over-expression? They mention that trypsine is synergized the SOD hub through CuZnSOD, but how does that work on the biochemical point of view? What about other important anti-oxidant proteins such as catalase of GSH peroxidase? What is the differential between trypsine and chitosan treatment?

Response: Yes, the recognition mechanism of trypsin is still unclear. It is hypothesized that trypsin activates the hormone signal pathway to regulate endogenous antioxidant system of H. undatus. We are going to check it.

As for the peroxidase, including GPX, APX, and so on, they are also complex works. A total of 101 peroxidase unigenes have been screened by us. The regulation and interaction of them are going to be investigated.

Chitosan was used as a positive control in the current work. The difference of weight loss between the two preservative groups (trypsin and chitosan) was not significant. While the effect of trypsin on the cell injury was much better than that of chitosan.

Overall, the article needs to be greatly improved before it could be considered for a publication.

If you have any question on this manuscript, please no hesitate to contact me at any time. Thank you very much!

Prof. Xin Li

College of Food and Bioengineering

Henan University of Science and Technology

Reviewer 3 Report

Comments on the manuscript “Transcriptomic Analysis Reveals hub Genes and Sub-Networks of SODs in Hylocereus undatus through Novel Superoxide Scavenger Trypsin Treatment during Storage”. Manuscript ID antioxidants-699062. Antioxidants.

The aim of the manuscript was to investigate the regulatory mechanisms induced by trypsin treatment on the Hylocereus undatus by cooperating with SODs. Results show that the trypsin treatment induced antioxidant activities or metal ion transport, while reduced the defense responses.

The goal of the work as well as the results are very interesting, however the article is not presented in a clear fashion and often it’s difficult to follow. English language should be improve. Anyway, results were interpreted appropriately in the discussion section and I think that it could provide an advance in current scientific knowledge. Experiments, statistics and data analyses were performed to a good technical standard. Nevertheless, a wide revision of the whole manuscript could greatly improve the quality of the paper and it's necessary before pucclication. I suggest to change the title in order to make it more attractive. The introduction is very poor and authors, in this section, should describe the background as well as the main knowledges which inspired their work. The section Materials and Methods is presented in a confused form including many short subsection, which often are missing of the main information to repeat an experiment, therefore authors must rewrite this section. Why did authors not measured the enzymatic activities of SOD, POD and lipoxygenase? The Results section contains a lot of results that need to be organized in a better form. For examples the table containing the primer sequences could be provided as supplementary material, however the results of validation by RT-qPCR should be displayed as main result.

Author Response

List of Responses

Dear editors and reviewers,

We have made some corrections based on the comments of you. I have highlighted the changes I make in the manuscript by using the track changes mode in Word. The responses to the comments of you are as follows:

Reviewer 3

Comments on the manuscript “Transcriptomic Analysis Reveals hub Genes and Sub-Networks of SODs in Hylocereus undatus through Novel Superoxide Scavenger Trypsin Treatment during Storage”. Manuscript ID antioxidants-699062. Antioxidants.

The aim of the manuscript was to investigate the regulatory mechanisms induced by trypsin treatment on the Hylocereus undatus by cooperating with SODs. Results show that the trypsin treatment induced antioxidant activities or metal ion transport, while reduced the defense responses.

The goal of the work as well as the results are very interesting, however the article is not presented in a clear fashion and often it’s difficult to follow. English language should be improve. Anyway, results were interpreted appropriately in the discussion section and I think that it could provide an advance in current scientific knowledge. Experiments, statistics and data analyses were performed to a good technical standard. Nevertheless, a wide revision of the whole manuscript could greatly improve the quality of the paper and it's necessary before pucclication. I suggest to change the title in order to make it more attractive. The introduction is very poor and authors, in this section, should describe the background as well as the main knowledges which inspired their work. The section Materials and Methods is presented in a confused form including many short subsection, which often are missing of the main information to repeat an experiment, therefore authors must rewrite this section. Why did authors not measured the enzymatic activities of SOD, POD and lipoxygenase? The Results section contains a lot of results that need to be organized in a better form. For examples the table containing the primer sequences could be provided as supplementary material, however the results of validation by RT-qPCR should be displayed as main result.

Response: Thanks for your kindly comments. The title and introduction have been revised. Some experiments have been used in our previous paper (Food & Function, 2019). So, these contents in Materials and Methods section are brief. The activities of antioxidant enzymes have been investigated in the paper published recently (Food & Function, 2019), and transcriptomic data was consisting with the enzyme activities determination. So, the enzymatic activities were not detected in the current work. Finally, the table 4 has been deleted and be renumbered as Table S12. Figure S14 has been changed to be Figure 7.

If you have any question on this manuscript, please no hesitate to contact me at any time. Thank you very much!

Prof. Xin Li

College of Food and Bioengineering

Henan University of Science and Technology

Round 2

Reviewer 1 Report

In the present form, the Referee can recommend for publication this manuscript in Antioxidants.

Author Response

List of Responses

Dear reviewer,

The responses to the comments of you are as follows:

Comments: 

In the present form, the Referee can recommend for publication this manuscript in Antioxidants.

Response: Thank you for your kindly recommendation.

If you have any question on this manuscript, please no hesitate to contact me at any time. Thank you very much!

Prof. Xin Li

College of Food and Bioengineering

Henan University of Science and Technology

Reviewer 3 Report

I think that authors improved their manuscript, although I think that many points of my previous revision were not addressed. Anyway, I think that the article is very interesting and deserve publication.

Author Response

List of Responses

Dear reviewer,

The responses to the comments of you are as follows:

Comments:

I think that authors improved their manuscript, although I think that many points of my previous revision were not addressed. Anyway, I think that the article is very interesting and deserve publication.

Response: Thanks for your kindly recommendation.

If you have any question on this manuscript, please no hesitate to contact me at any time. Thank you very much!

Prof. Xin Li

College of Food and Bioengineering

Henan University of Science and Technology

This manuscript is a resubmission of an earlier submission. The following is a list of the peer review reports and author responses from that submission.

Round 1

Reviewer 1 Report

The manuscript describes transcriptomic analysis of pitaya fruits during storage after treatment with trypsin, a novel scavenger of superoxide anions and identification of hub genes and subnetworks of SODs in pitaya.

This manuscript contains new information on the effect of trypsin treatment as a scavenger of superoxide anion in the preservation of pitaya fruit and the changes in its transcriptome, and also provides promising information for application.

Author Response

Thanks for your support.

Reviewer 2 Report

The manuscript describes a work about transcriptomic analysis reveals hub genes and sub3 networks of SODs in Hylocereus undatus through
4 novel superoxide scavenger trypsin treatment during
5 storage. The whole manuscript needs to improve mainly the introduction, the experimental part, and results. The figures must be changed for a better presentation, in my opinion after this overall improvement the manuscript could be accepted. 

Author Response

The manuscript describes a work about transcriptomic analysis reveals hub genes and sub3 networks of SODs in Hylocereus undatus through novel superoxide scavenger trypsin treatment during storage. The whole manuscript needs to improve mainly the introduction, the experimental part, and results. The figures must be changed for a better presentation, in my opinion after this overall improvement the manuscript could be accepted.

Response: The description of pitaya has been expanded in the introduction part. The corresponding reference has been revised. The descriptions have also been revised in the methods and discussion part.

Reviewer 3 Report

The authors of the manuscript “Transcriptomic analysis reveals hub genes and sub networks of SODs in Hylocereus undatus through novel superoxide scavenger trypsin treatment during storage” described the use of trypsin as superoxide scavenger. It is interesting and hot topic the use of storage preservatives in the food industry, having in mind that the best preservative is low temperatures.

The authors in the introduction fails in clarify the rational and the aims are not clearly defined. The authors cannot generalize trypsin use because each organism contain different enzymatic content. The approach must be interpreted as which enzymes are disrupt with trypsin, in the particular system of pitaya at 25 C. The transcriptomic analysis is well-done, however bad merge in the story.

The statement in the conclusion at least is over-projected, based on what trypsin can control the fruit storage. What about the taste? The chemical composition, the nutritional value, varieties, chemotypes, etc?  The authors did not mentioned several other factors, which affect the fruit quality not only the visual one.

How trypsin disrupt protein-protein interaction, please expand.

The authors must be focus in a better to say the story more flow the manuscript and it will be a good contribution.

Minor correction.

I noticed several font size through the paper for example at the abstract line 25 different than 24.

In the introduction, the authors must state why they study pitaya. Is there any economic impact? The description of the importance of pitaya is short please expand.

Some notation are wrong H2O2? Page 5 line 169.

At this point, the manuscript raised several questions that must be address before consideration to publish.

Author Response

Dear Prof. Rebecca Shu,

We have made some corrections based on comments of editors. I have highlighted the changes I make in the manuscript by using the track changes mode in Word. The response to the comments of reviewer 3 are as follow:

Comments and Suggestions for Authors

The authors of the manuscript “Transcriptomic analysis reveals hub genes and sub networks of SODs in Hylocereus undatus through novel superoxide scavenger trypsin treatment during storage” described the use of trypsin as superoxide scavenger. It is interesting and hot topic the use of storage preservatives in the food industry, having in mind that the best preservative is low temperatures.

The authors in the introduction fails in clarify the rational and the aims are not clearly defined. The authors cannot generalize trypsin use because each organism contain different enzymatic content. The approach must be interpreted as which enzymes are disrupt with trypsin, in the particular system of pitaya at 25 C. The transcriptomic analysis is well-done, however bad merge in the story.

Response: Sure, even though all plants use antioxidant enzymes including CAT, SOD, POD, APX and so on, to protect cells from damage of excess ROS. The species, activities and interacting mechanisms of endogenous enzymes in different fruits will be different. The storage temperature indeed impact on the metabolisms of fruits during storage. As a result, storage temperature could affect on the shelf life of fruits. While, the interaction patterns between enzymes will not change. In this current study, experiments were performed at 25 °C, which are approximate to the annual room temperature. Since trypsin acts as a superoxide scavenger with fruits during storage, antioxidant enzymes are impacted. As shown in this manuscript, 7 SODs were up- or downregulated by trypsin. Results revealed that important physiological metabolisms, such as antioxidant activities or metal ion transport, were induced, while defense responses were impeded by trypsin. Illustration of the SODs regulated by trypsin should lead us a new view on the preservation mechanism of pitaya and reveal new insights into regulatory mechanisms of other antioxidant enzymes and even other fruits.

The statement in the conclusion at least is over-projected, based on what trypsin can control the fruit storage. What about the taste? The chemical composition, the nutritional value, varieties, chemotypes, etc?  The authors did not mentioned several other factors, which affect the fruit quality not only the visual one.

Response: Thanks for your kindly suggestion. The descriptions in the discussion (line 322) and conclusion (line 401) were revised. In this manuscript, we try to talk about the cell protection of trypsin. So, cell damage related indexes, including MDA contents, superoxide production and H2O2concentration were detected. The factors including taste, chemical composition, the nutritional value, etc. did not be obtained. In the next manuscripts, we’d like to complete the determination of indexes and further discuss the mechanisms of fruit quality improvement by trypsin.

How trypsin disrupt protein-protein interaction, please expand.

Response: In the current study, the hub genes and sub-networks of SODs in Hylocereus undatus through novel superoxide scavenger trypsin treatment during storage were investigated. While, how trypsin performs this function is unclear yet. The receptors, signals and pathways of trypsin still need more work. We are going to work on the further mechanisms.

The authors must be focus in a better to say the story more flow the manuscript and it will be a good contribution.

Response: The description has been revised in the whole text, especially in discussion part.

Minor correction.

I noticed several font size through the paper for example at the abstract line 25 different than 24.

Response: The style of font has been corrected.

In the introduction, the authors must state why they study pitaya. Is there any economic impact? The description of the importance of pitaya is short please expand.

Response: The description of pitaya has been expanded. The corresponding reference has been revised.

Some notation are wrong H2O2? Page 5 line 169.

Response: The style of H2O2has been corrected.

At this point, the manuscript raised several questions that must be address before consideration to publish.

If you have any question on this manuscript, please contact me at any time. Thank you very much!

Prof. Xin Li

Round 2

Reviewer 2 Report

The manuscript describes a work about a transcriptomic analysisthat reveals hub genes and sub networks of SODs in Hylocereus undatus through novel superoxide scavenger trypsin treatment during storage.

In the abstract do not start with:” It was demonstrated in our previous research that trypsin scavenges superoxide anions.” This could be explained in the introduction and included indirectly in the beginning of the abstract:

Start with “In this study, the synergistic mechanisms of trypsin and superoxide dismutases (SODs) were 22 evaluated in Hylocereus undatus (pitaya). …

All Figures must be improved (are too small) because is hard to see the information.

The remaining changes improved the work and so after the slight modifications the paper could be accepted.

Author Response

The manuscript describes a work about a transcriptomic analysisthat reveals hub genes and sub networks of SODs in Hylocereus undatus through novel superoxide scavenger trypsin treatment during storage.

In the abstract do not start with:” It was demonstrated in our previous research that trypsin scavenges superoxide anions.” This could be explained in the introduction and included indirectly in the beginning of the abstract:

Start with “In this study, the synergistic mechanisms of trypsin and superoxide dismutases (SODs) were 22 evaluated in Hylocereus undatus (pitaya). …

Response: The sentence “It was demonstrated in our previous research that trypsin scavenges superoxide anions.” described the key background of this manuscript. I prefer to keep it.

All Figures must be improved (are too small) because is hard to see the information.

Response: Yes, the figures of PPI networks were not clear enough. Node sizes of figure 5 and figure 6 have been improved. While, the purpose of figure 4 is to show the morphology and clusters of entire network of 447 nodes. The full information has been listed at table S5.

The remaining changes improved the work and so after the slight modifications the paper could be accepted.

Reviewer 3 Report

The explanation from the authors address my comments 

Author Response

Thank you so much!